# Neural network-based symmetric encryption algorithm with encrypted traffic protocol identification

Jiakai Hao[1], Ming Jin[1], Yuting Li[1] and Yuxin Yang[2]

[1] Information and Communication Branch, State Grid Beijing Electric Power Company, Beijing, China

[2] School of Computer Science, Beijing University of Technology, Beijing, China

## ABSTRACT

Cryptography is a cornerstone of power grid security, with the symmetry and asymmetry of cryptographic algorithms directly influencing the resilience of power systems against cyberattacks. Cryptographic algorithm identification, a critical component of cryptanalysis, is pivotal to assessing algorithm security and hinges on the core characteristics of symmetric and asymmetric encryption methods. A key challenge lies in discerning subtle spatial distribution patterns within ciphertext data to infer the underlying cryptographic algorithms, which is essential for ensuring the communication security of power systems. In this study, we first introduce a plaintext guessing model (SCGM model) based on symmetric encryption algorithms, leveraging the strengths of convolutional neural networks to evaluate the plaintext guessing capabilities of four symmetric encryption algorithms. This model is assessed for its learning efficacy and practical applicability. We investigate protocol identification for encrypted traffic data, proposing a novel scheme that integrates temporal and spatial features. Special emphasis is placed on the performance of algorithms within both symmetric and asymmetric frameworks. Experimental results demonstrate the effectiveness of our proposed scheme, highlighting its potential for enhancing power grid security.

## INTRODUCTION

With the rapid development of information technology, the importance of information security has become increasingly prominent. Every aspect of modern society relies on information technology, from personal communication and financial transactions to national security, which requires secure information systems to ensure data confidentiality, integrity, and availability. However, the advancement of information technology has also brought challenges to information security. Cyberattack methods are becoming increasingly complex and diverse, with threats such as data breaches, identity theft, malware, and distributed denial-of-service attacks on the rise, posing serious security risks to individuals, enterprises, and nations (*Hayashi & Koshiba, 2023*). Encryption and data protection through cryptographic systems are essential to achieving information security. The cryptanalysis of ciphertexts is also an important branch of cryptography



Corresponding author
Yuxin Yang,
yangyangyang@emails.bjut.edu.cn

(*Koshiba, Zolfaghari & Bibak, 2023*). There are two main directions in cryptographic algorithm identification: to carry out reverse analysis of software programs and hardware systems and to identify cryptographic algorithms (*Zhang, Shu & Jiang, 2011*). For software programs, disassembling and dynamic debugging techniques are used to analyze the execution flow and functional structure of the code to deduce the cryptographic algorithm used in the program; for the analysis of hardware devices, the physical characteristics of electronic devices, such as side-channel detection technology and the design structure of electronic components, are used to deduce the cryptographic algorithm. Another area of research is identifying cryptographic algorithms to extract relevant features with the ciphertext in the ciphertext-only state and to deduce the algorithm to which the ciphertext belongs. Shannon proposed two crucial principles of obfuscation and diffusion in 1949, which are widely used in modern cryptographic algorithm design. Based on these principles, modern cryptographic algorithms are designed with complex network structures, polling mechanisms, or mathematical problems to hide shallow features as much as possible, enhance the difficulty of cracking, and try to make the cypher as similar to random numbers as possible (*Hayashi & Koshiba, 2022*). However, some researchers have found that some cryptographic algorithms generate ciphertext data between the randomness of the numerical differentiation problem. This provides a critical theory for the ciphertext-only state to distinguish between cryptographic algorithms (*De Souza & Tomlinson, 2013*). Extracting features from the ciphertext can determine and identify cryptographic algorithms without relying on the key or other auxiliary information.

In the ciphertext-only state, there are two main categories of cypher algorithm identification methods: one is based on statistics and the other is based on machine learning. With the increasingly complex design structure of cryptographic algorithms, the method based on statistics is no longer applicable, and its essence is to utilize statistical knowledge to design indexes and compare the results obtained from the ciphertext statistical computation with the preset thresholds to infer the type of cryptographic algorithm. With the development of artificial intelligence technology, machine learning methods are applied to cryptographic algorithm identification (*Gong, Yao & Nallanathan, 2024*). The recognition issue is regarded as a categorization challenge. A supervised modelling approach is employed to input this ciphertext feature training dataset into the classifier model to train. After the end of the course, and the test dataset is input into the classifier model to output the classification results. The deep learning method is a powerful branch of machine learning methods; However, using deep learning technology to study cryptographic algorithm recognition is still in its infancy. It stands out in the algorithm recognition task because of its excellent representation learning ability. It has gradually become the mainstream research at this stage (*Jung et al., 2024*). In the study of cryptographic algorithm recognition, ciphertext feature selection and analysis are essential tasks and cryptographic algorithm category recognition accuracy is still generally low. In the face of irregular ciphertext data, single-dimensional features can no longer adequately reflect the ciphertext information, so attempts are made to extract multi-dimensional ciphertext features and utilize deep learning algorithms to mine the underlying feature

relationships of ciphertexts to identify the ciphertexts belonging to the cryptographic algorithms (*Alwhbi, Zou & Alharbi, 2024*).

Securing network communications, protecting network protocols, and identifying network data stream techniques have been iteratively updated. Currently, the recognition methods based on port, deep packet inspection and artificial intelligence are no longer enough (*Fujita, Koshiba & Yasunaga, 2022*). Meanwhile, deep learning-based methods have been explored and utilized across various domains, and several researchers have implemented deep learning techniques for network traffic identification (*Altaf et al., 2024*). By reviewing and analyzing the literature on network traffic identification, it has been found that this field primarily focuses on three areas (*Zhang et al., 2022*): identification of the application of encrypted traffic, identification of the protocol of encrypted traffic and detection of illicit traffic on the network. There is no fundamental distinction in the research among these areas; they primarily concentrate on recognition schemes and the choice of traffic characteristics. Significant advancements in these areas will drive development and progress in the field. The current state of research can be introduced from the viewpoint of extracting traffic characteristics; it can be categorized into the following aspects: spatial traffic attributes, temporal traffic attributes, and combined temporal-spatial traffic attributes (*Alrayes et al., 2023*).

The contributions of our work can be described as follows.

(1) We propose a convolutional neural network (CNN) model for plaintext guessing of symmetric cryptography algorithms: Symmetric cryptography guessing model (SCGM), and try to guess the plaintexts of four symmetric cryptography algorithms to evaluate the model's learning ability and practicality.

(2) We study the recognition task of encrypted traffic protocols, optimize the effect of communication data recognition through data preprocessing and session-level processing, and propose an algorithmic recognition scheme that incorporates spatio-temporal features to improve recognition accuracy.

(3) Finally, the validity of the SCGM model and the usability of the algorithmic identification scheme incorporating spatiotemporal features are demonstrated experimentally.

## RELATED WORK

### Cryptographic algorithm recognition

In 2015, *Hongchao (2018)* used NIST's randomness test for the first time to extract ciphertext features and classify and recognize five grouped cryptographic algorithms and achieved recognition accuracy of about 70% for two-by-two clustering using the K-means algorithm. This study further explores the development of ciphertext randomness metric features in cryptographic algorithm recognition. In 2018, *Liangtao, Zhicheng & Yaqun (2018)* formally defined the cryptographic algorithm recognition system for the first time, laying a theoretical foundation for standardizing the cryptographic algorithm recognition problem into two aspects: the design of the recognition scheme and the extraction of ciphertext features. On this basis, the random forest algorithm is used as the recognition model in the study, and three types of recognition scenarios are set up for four

cryptographic regimes: classical cypher, stream cypher, group cypher and public key cipher. At the same time, a layered recognition scheme is proposed, which includes two recognition phases: cluster partition and single partition. Compared with the single-layer recognition scheme studied by previous researchers, the recognition accuracy of the layered recognition scheme is improved by about 20%.

In 2020, *Ma et al. (2024)* improved the theoretical framework of the recognition problem of cryptographic algorithms. Combined with the Relief feature selection algorithm, a dynamic scenario recognition scheme for cryptographic algorithms based on integrated learning was proposed, and the optimal recognition accuracies using this method were improved by 6.41%, 10.03%, and 11.40%, respectively, under the three recognition scenarios in the study. In 2021, *Huang (2021)* proposed three hybrid models based on the integrated learning idea and used the NIST randomness test. The National Institute of Standards and Technology (NIST) randomness test method is used as the feature extraction method, and in the single-layer recognition task of cryptographic algorithms, the two-classification and five-classification recognition for five grouped cryptographic algorithms, namely, Advanced Encryption Standard (AES), 3 Data Encryption Standard (3DES), Blowfish, Carlisle Adams and Stafford Tavares (CAST), and Rivest Cipher 2 (RC2), are accomplished, respectively. Among them, the accuracy of AES and 3DES binary classification recognition is not less than 70%; in the binary classification recognition task, the highest accuracy can reach 77.5%; the most accurate among the five classification results is 38%, and this recognition effect is better than the non-integrated learning algorithms. In their studies, *Liru (2021)*, *Dai et al. (2022)*, *Lin et al. (2024)* extracted the features of the ciphertexts by using the NIST randomness test method, which was the first time the deep learning model was applied to the field of cryptographic algorithm recognition. The study used a random forest algorithm, backpropagation (BP) neural network, convolutional neural network and sequential neural network algorithms to train the corresponding cryptographic algorithm recognition classifiers to recognize eight cryptographic algorithms. The experimental findings demonstrate that deep learning-based encryption algorithm recognition systems outperform machine learning recognition schemes such as SVM and random forest, and the recognition accuracy is about 30% higher than that of the random forest-based scheme. The research in *Chuxuan (2021)*, *Li et al. (2024)*, *Ariyanto et al. (2023)* proposed a ciphertext feature extraction method based on linear transformation and convolutional sampling. In the article, the cumulative sum method is utilized to process the binary sequences of ciphertext files and used as ciphertext features to construct a random forest classifier to identify two-class classification and multi-class classification of four cryptosystems such as AES, 3DES, Blowfish, and RSA. The experimental results confirm the high feasibility of ciphertext throttling sequence features in cryptographic algorithm recognition research.

## Network traffic analysis

The researchers in *Luyu, Liao & Zhao (2018)* and *Ding et al. (2024)* employed a convolutional neural network with three convolutional layers to extract the initial 1,024

bytes of network packet data, converting them into grayscale images. They also used one-hot encoding for the labels to train the model. The empirical findings indicate that the model attains an average recognition rate of over 90% for Transmission Control Protocol (TCP), User Datagram Protocol (UDP), and unknown protocols. However, the method is less effective in recognizing software layer protocols of secure communication traffic and lacks universality. In 2019, *Wang Lin & Biao (2019)* introduced a genetic algorithm-based improvement to the random forest model, focusing on time-series network flow characteristics. This method demonstrates improved recognition compared to directly using the random forest algorithm, achieving around 92% accuracy in identifying 14 SSL protocol applications. *Jing (2019)* performed a reconnaissance study on seven types of encrypted application traffic. They built a deep neural network model and conducted experimental comparisons with the basic Bayesian algorithm. The study emphasized the impact of mini-batch size on the model's training effectiveness. Results revealed that using a value of 40 yielded the highest recognition accuracy, significantly surpassing the recognition capability of the basic Bayesian algorithm. The study in *Zhao (2022)* employed a temporal examination of network data and suggested a recognition methodology. That combines time-based characteristics with a support vector machine (SVM). By segmenting into equal-length sequences at the session level, autonomously learning temporal features using the Shapelet-Transformer algorithm, and building classifiers with SVM, the model achieves a 95% accuracy rate in recognizing normal and malicious traffic.

Nevertheless, this method's recognition efficiency falls short of expectations. In 2021, *Jisheng, Zheng & Sweet (2022)* delved into application layer protocol recognition using the ISCX2012 dataset. They extracted spatial features through residual networks and temporal features through bidirectional gated recurrent unit network (BiGRU). The model achieved an overall accuracy of 96.87%, yet it did not extensively investigate the capability for recognizing encryption protocols.

## METHOD

Firstly, we propose a model for symmetric cryptography algorithms to perform plaintext guessing: the SCGM, which is a combined convolutional neural network model, and the model to try to think the plaintexts of the four symmetric cryptography algorithms. The strength of the model's learning ability and the practicality of this method are judged by the results shown in the plaintext data of the four symmetric cryptography algorithms that are guessed.

Second, we focused on protocol identification of encrypted traffic data and preprocessed the data accordingly. We analyzed the variability among the captured network traffic and employed session-level processing to ensure the best communication data recognition results. Meanwhile, we deeply analyzed the influence of single features among temporal and spatial features on the recognition effect. We proposed an algorithmic recognition scheme that fuses temporal and spatial features of encrypted traffic protocols.

## A neural network model-based scheme for symmetric ciphertext plaintext guessing

In this article, we propose a model based on a CNN called the SCGM. We utilize TensorFlow as the foundational framework to predict four symmetric encryption algorithms: AES, Data Encryption Standard (DES), Blowfish (Symmetric Encryption Block), and RC4 (Stream Encryption Algorithm).

The decision to employ a CNN is driven by its strong local perception capability, particularly relevant given the spatial distribution of plaintext and ciphertext. Each node in the information transfer process must maintain integrity and facilitate smooth transmission, making the interconnections between characters crucial. The spatial mapping ability of CNNs allows for effective local feature extraction, enabling the model to capture relationships within the data efficiently. Additionally, the translation invariance of CNNs ensures stability in spatial matrices, allowing for robust convolutional feature extraction while preserving the structural integrity of the information. These properties make CNNs well-suited for constructing a model that effectively analyzes encryption patterns.

The network can solve spatial problems by combining multi-module convolutional neural networks. Convolutional neural networks can abstract an image as a two-dimensional matrix of pixels and use this matrix to perform operations such as convolution and pooling, thus improving the efficiency and accuracy of image processing. A convolutional neural network model usually consists of multiple layers, including an input layer, a convolutional layer, a pooling layer, a fully connected layer, and an output layer. The number and order of these layers can be adjusted and optimized according to the application's needs.

A convolutional neural network is an efficient image-processing technique that extracts features through convolution and pooling operations and can process image data efficiently with high classification and recognition accuracy. According to Eq. (1), the input of the convolution operation is x(t), the kernel function is k(t), and the output is y(t).

$$H^j = f\left(b^j + \sum_i w^{ij} * x^i\right). \tag{1}$$

Two fundamental concepts in convolutional operations, local connectivity and parameter sharing, play a huge role in convolutional operations. Local connectivity: 'Local connectivity' means that neurons in a neural network are not entirely connected but are only adjacent. 'Weight sharing' means sharing a weight between multiple neurons in a neural network. This approach reduces the number of parameters, reduces computational complexity, improves the network's performance, and enables regularisation. With weight sharing, the number of parameters can be significantly reduced as they can use fewer parameters with the same weights to effectively express the same information, thus considerably improving the generalization ability of the neural network.

The  Natural Language Toolkit (NLTK) is a Python library that supports and simplifies natural language processing. The character-level technique is a natural language

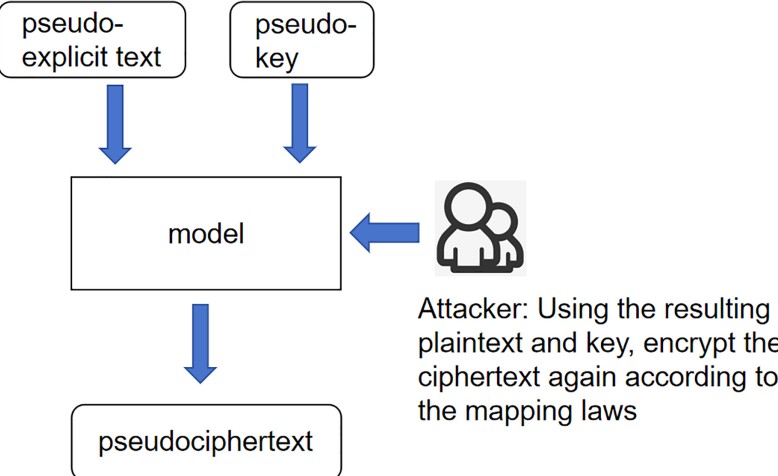

**Figure 1  Scenarios with unknown keys.**

processing technique that breaks down text into individual characters by first collecting 57,000 sentences in the Brown English *corpus* and then splitting each word in each paragraph of the sentence, thus enabling the encryption of the word or sentence and analyzing and processing based on these characters. To better understand the plain text, the 57 most common characters were chosen and the 'unknown' mark was added to distinguish the unencrypted words in the plain text. After encryption by the encryption algorithm, a list of 85 words was reconstructed to better understand each word in the plain text.

In data processing, sentences must be extracted from the NLTK library, and their lengths must be truncated. If the size of the sentences exceeds a predefined range, they are truncated, while if the size of the sentences is less than a predefined range, a zero-completion operation is performed. The original plaintext can be converted into a cypher and digitally encoded using encryption algorithms. Finally, the encoded cypher can be processed to reduce the computational complexity of the neural network matrix.

This article will describe the following three scenarios: unknown key scenario, unknown plaintext scenario, and unknown ciphertext scenario. Unknown key scenario: This scenario's core meaning is that the key encrypts the plaintext to generate the ciphertext. Then, according to the refinement of this process, it can be considered that the interaction between the plaintext and the key generates the ciphertext. The interaction between the plaintext and the ciphertext generates the ciphertext, which is not necessarily the key. The process is not a simple role, but the plaintext and the ciphertext can generate new roles between the plaintext and the ciphertext, where the role generator is called a pseudo key. Here, this interaction is called a pseudo-key. This overall scenario is 247, called the unknown critical scenario as shown in Fig. 1.

Figure 1 illustrates the interactions between the pseudo-explicit text, pseudo-key, and the model. The attacker utilizes the resulting plaintext and key to re-encrypt the ciphertext based on the mapping laws. This process is central to the model's ability to infer plaintext

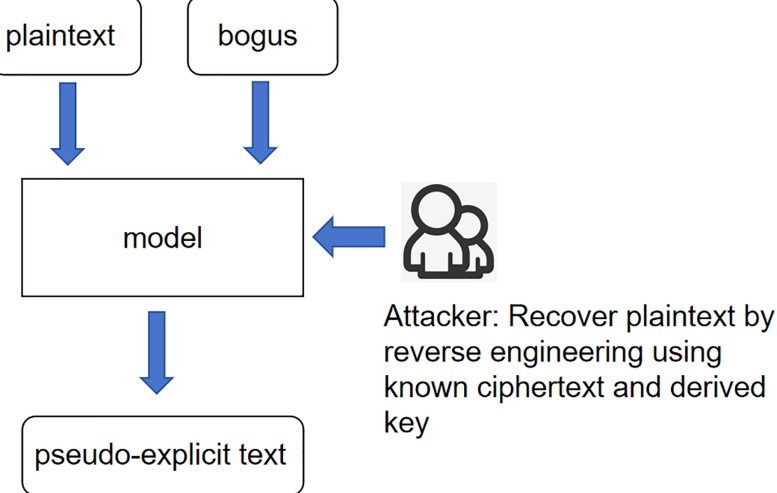

**Figure 2 Scenarios with unknown plaintext.**

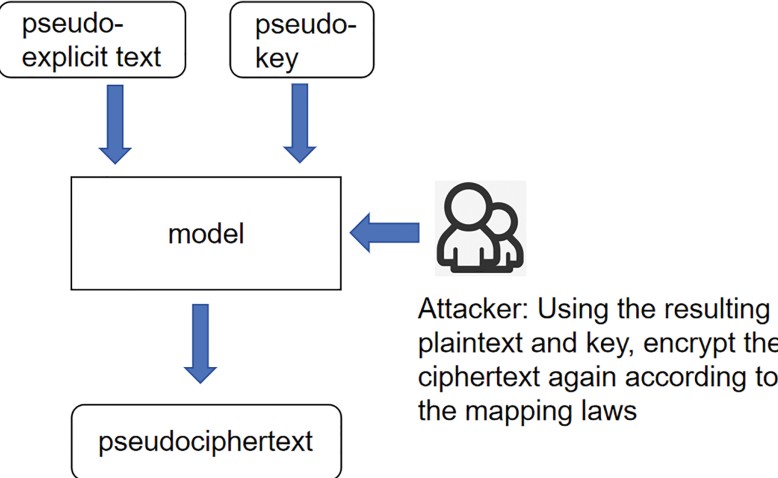

**Figure 3 Scenarios with unknown ciphertext.**

from ciphertext under scenarios where the key is unknown. Unknown plaintext scenario: The core meaning of this scenario is still the same as 249 in the previous scenario, except that the input variables have been changed to ciphertext 250 and pseudo-key, and according to the expression of the last idea, the product of the 251 interaction between ciphertext and pseudo-key is called pseudo-plaintext. This whole 252 scenario is called the unknown plaintext scenario as shown in Fig. 2.

Unknown ciphertext scenario: the core meaning of this scenario is the same as the expression of the previous two scenarios; the input variables are changed again to pseudo-plaintexts and pseudo-keys, and according to the idea stated earlier, then the product of 256, the interaction between pseudo-plaintexts and pseudo-keys is called pseudo-ciphertexts. And call this whole scenario an unknown ciphertext scenario, as shown in Fig. 3.

After the above three scenarios are constructed, tasks can be assigned to each convolutional neural network module, and the parameters of the input layer, convolutional layer, fully connected layer, and output layer of the convolutional neural network can be set. The module coupling is also adjusted to make the data reasonable and representative. It is hoped that the data can be guessed through the different scenarios without considering the changes in other variables. With further adjustments, the model may be compatible with asymmetric encryption systems, which include RSA and elliptic curve cryptography (ECC). The system needs specific modifications because encryption schemes and key management systems differ between different implementations. Extra research and specialized feature extraction techniques for asymmetric encryption structures must be developed to enable the model to generalize between symmetric and asymmetric techniques. The development of an expanded dataset for analyzing abilities and adaptation modifications to the model remains vital because it aims to include AV, encryption techniques, and their unique characteristics. Expanding and adapting the model remains important because the methodology analyzes different encryption schemes while creating new methods to extract better-modifying standards.

## Algorithmic fusion of spatiotemporal features for encrypted traffic protocol identification

We suggest an encrypted traffic protocol recognition scheme that merges spatiotemporal traffic features, employing Transformer and attention CNN models to handle input data. The model identifies encrypted traffic protocols using the Transformer encoder model to process network data's temporal byte stream details and the CNN model with an attention mechanism to process its spatial information. The CNN model maintains the conventional structure of convolutional neural networks with layers such as convolutional, pooling, and fully connected layers. Additionally, it incorporates a normalization layer to enhance generalization capability. The model inputs preprocessed session-level traffic classification data and generates predicted object labels as output. Including an attention mechanism improves the distinction among network protocol features, facilitating the accurate identification of network protocols.

Considering the timeliness of the recognition in massive network traffic data, the two models in the recognition scheme adopt a parallel structure. The parallel structure also has the following advantages: (1) Since different forms of data sources have different features and structures, the use of different models can better extract and process the features of the data sources, thus improving the recognition accuracy of the neural network; (2) The use of the parallel structure can separately process and extract the features of the different data sources, thus reducing the training time of the model and the demand for computational resources; (3) The scalability of the neural network can be improved, and more models can be added to handle different types of data sources, thus improving the recognition ability of the neural network.

Network traffic data consists of bits, which can be viewed as hexadecimal sequences or strings for ease of understanding and is essentially one-dimensional data. The traffic data is first preprocessed into images, and the one-dimensional data is converted into

two-dimensional 28*28 data. In the first convolution layer to train a two-dimensional convolution kernel, traffic features grayscale map input to the CNN model, apply a 16-channel $3 \times 3$ convolutional kernel, resulting in a feature map with 16 channels. With appropriate padding parameters, the feature map size remains at $28 \times 28$. The convolution operation is expressed by Eq. (1):

$$H^j = f\left(b^j + \sum_i w^{ij} * x^i\right). \tag{2}$$

$H^j$ and $x^i$ i denote the jth output mapping and the ith input mapping, respectively. $w^{ij}$ represents the convolution filter weights, * signifies convolution, and $b^j$ is the bias component of the jth mapping. The function f signifies the transfer function, with ReLU being utilized in this convolutional layer to bolster the model's resilience against over-learning. The transfer function is denoted by Eq. (2) as

$$f(x) = \begin{cases} x, & x > 0 \\ 0, & x \leq 0 \end{cases}. \tag{3}$$

We transformed the single-dimensional traffic data by dividing its raw data into specific-length sequences prior to image conversion. The resolution of $28 \times 28$ emerged from experimental selection because it struck a dependable compromise between model efficiency and fundamental temporal and spatial element retention from the raw data. Due to the $28 \times 28$ dimensions, the model maintains a suitable input size for its convolutional neural network component while capturing data spatial relationships locally. Padding is used to make every sequence uniform in length. The data sequence needs extra zero-padding sent to its end to reach 784 pixels (28 pixels by 28 pixels). Long sequences with more than 784 elements will be cut down to match the dimension $28 \times 28$. The method achieves data normalization across inputs while reducing the computational workload. The data is normalized into a grayscale intensity range from 0 to 255 to make the input features usable by the model. By combining padding, scaling, and normalization, the model's processing efficiency increases, as does its ability to discover relevant features. ReLU helps the model explore feature correlations better, enhance fitting capabilities, and mitigate gradient vanishing issues. A batch normalization layer is included before the activation function to normalize features, ensuring a zero-mean state and a statistical dispersion of one.

The model includes three convolutional layers, three pooling layers, a normalization layer, an attention mechanism, and a fully connected layer. The final output feature vector is linked to the attention layer; the output from the attention layer is then fed into a fully connected layer consisting of 128 neurons, incorporating dropout for regularization. The activation mechanism employed is ReLU. Following the activation mechanism, the output is combined with the ultimate result from the Transformer encoding module.

Figure 4 illustrates the flowchart of spatio-temporal feature fusion for recognizing types of encrypted traffic protocols. During training, starting from the training data Pcap file, the

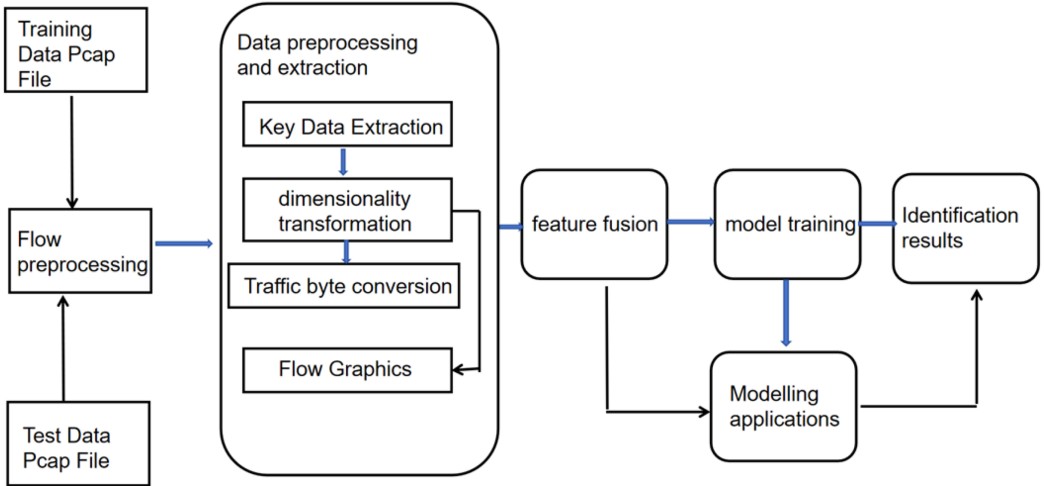

**Figure 4** Flowchart for encrypted traffic protocol recognition with spatio-temporal feature fusion.

data file undergoes the stages of preprocessing, Feature extraction and fusion, model training, outputting recognition outcomes, and ultimately, the protocol recognition model trained; during training, the same data Pcap file chosen, and the selected file undergoes stages of preprocessing, feature extraction, and feature fusion, and then the file is inputted into the trained model for recognition of traffic protocols. The final output is the recognition results.

# EXPERIMENT

## Experimental environment

The experiments utilized version 1.7.0 of PyTorch through PyCharm as their development environment. NVIDIA RTX 3090 GPU and 43 GB of RAM operated with an Intel Xeon Platinum 8350C CPU (2.60 GHz). Ubuntu 18.04 was used as the operating system, and the Python version was 3.8. The dataset employed for training consisted of more than 100,000 labelled examples of encrypted network data. The training process required 50 epochs, with each batch containing 64 samples. Each model required various amounts of time for training, and the total training period reached 6 h based on the encryption algorithm's intricacy and the model design specifications. The experiment uses PyTorch deep learning framework and PyCharm software as the 320 model training environment; the specific hardware and software environment is shown 321 below:

The traffic data contains different cryptographic system patterns that represent standard encryption algorithms. A distinct dataset containing various encrypted network traffic for eight protocol types, such as Hypertext Transfer Protocol (HTTP), Domain Name System (DNS), UDP, and secure shell (SSH), served to develop the protocol identification scheme. The database we used for this operation includes unprocessed traffic information and annotations at the session level. The samples within traffic datasets contain three classes of information, including byte stream records, packet measurement data, and timestamp measurements. The training data included more than 100,000

**Table 1 Experimental hardware and software environment.**

| Equipment name | Equipment model |
| --- | --- |
| Display card (computer) | RTX3090 |
| RAM | 43G |
| CPU | Intel(R)Xeon(R)Platinum 8350 C CPU 2.60 GHz |
| Operating system | Ubuntu 18.04 |
| Python | 3.8 |
| PyTorch | 1.7.0 |

instances of encrypted traffic that received processing and then transformed into feature vectors for model input. The data split divided the information into two parts: training alongside testing, where 70% of elements were used for training and 30% for testing. Different algorithms have different encryption processes, and the length of the encryption parameters is also different. Therefore, the plaintext length, key length, and ciphertext length of the inputs of the AES algorithm, DES algorithm, Blowfish algorithm, and RC algorithm are designed in the following way, as shown in the Table 1. The encryption algorithm depends on the relationship between plaintext, key, and ciphertext length. Usually, the ciphertext length is fixed, while the key length can affect the security and performance of the encryption algorithm. This article focuses on a fixed length for analysis. Symmetric algorithm data containing AES, DES, Blowfish, and RC4 were used during the experimental training phase for the SCGM model. The training consisted of pairs containing ciphertext and plaintext data, which were used to create a prediction model that derived plaintext from ciphertext.

The evaluation metrics for symmetric algorithm performance measured accuracy with precision, recall, and F1-score. The accuracy assessments counted correct plaintext predictions to the entire sample collection. The precision evaluation consisted of two elements: the adequate prediction of positive plaintext results and the total predictions of positive plaintext. The ratio between correctly predicted positive plaintext samples and the total actual positive samples represented the measure for recall performance. The F1-score function calculated as the precision and recall harmonic mean offered a performance balance for the model. The SCGM model did not directly evaluate asymmetric algorithms, but the investigation examined protocol identification that could require asymmetric encryption. Similar metrics evaluated the protocol identification scheme through accuracy, precision, recall, and F1-score measurements to measure classification success. Our model evaluation included utilizing the area under the curve method from receiver operating characteristic curves to measure its effectiveness in differentiating between different protocols as in Table 2.

## Experimental results and analysis

The experimental dataset consists of encrypted traffic generated through four symmetric encryption algorithms: AES, DES, Blowfish, and RC4. This dataset enables model evaluation and comparison but fails to deploy real-world traffic encryption variety since it

**Table 2 Setting the length of plaintext, key, and ciphertext for different algorithms.**

| Arithmetic | Explicit length | Key length | Ciphertext length |
|---|---|---|---|
| AES-128 | 128 | 128 | 128 |
| DES | 64 | 64 | 64 |
| Blow-fish | 128 | 64 | 128 |
| RC4 | 128 | 64 | 128 |

excludes complex or combination encryption systems and protocols not analyzed in this study. The synthetic dataset lacks endpoint features that characterize real-world encrypted traffic because the latter contains dynamic elements that do not exist in the former. Since other methods are based on experiments with passwords generated by specific 332 rules, such as HashCat Best64, HashCat gen2, and JTR SpiderLab, the passwords generated in 333 in this article are based on symmetric encryption algorithms so that no comparison can be made 334 here. The hit rate results of plaintext guessing of the AES algorithm are shown in Fig. 5. In the SCGM model, as the number of iterations increases, the learning ability of the 336 model also increases, so the hit rate curve under the model is upward. Then, the hit 337 rate is gradually stabilized at about 60%, which indicates that the SCGM model is adequate. The curve stabilizes at 60% after 50 epochs with a batch size of 64 and a learning rate of 0.001. The x-axis represents the number of training iterations (epochs), and the y-axis shows the accuracy achieved by the model on the plaintext guessing task. The SCGM model reaches a 60% success rate in plaintext prediction during a challenging encryption task, which prevents it from accessing the encryption secret. The achieved accuracy level through this model serves practical purposes because it proves its ability to uncover significant patterns inside encrypted information. Within cryptanalysis, the challenge of obtaining plaintext when lacking key access is reduced when the methodology delivers a 60% success rate. The 60% accuracy rating provides an initial point from which future model developments can be built utilizing better data inputs, model optimization, and advanced techniques.

The assessment includes experimental results and theoretical analysis of our model's operating performance. The SCGM model contains convolutional layers that perform local feature extraction that provides built-in resistance against minor encryption pattern fluctuations through the effective attainment of spatial relationships by filters. Through its self-attention mechanism, the Transformer boosts robustness by understanding distant acquaintances between elements in the dataset, thus ensuring accurate predictions despite troublesome input data. The model architecture possesses an efficient design for encrypted traffic variations by uniting temporal and spatial features to determine the most crucial protocol information. The model can generalize across encrypted datasets and encryption methods because of its adaptive design. To verify the effectiveness of the suggested Transformer-Attention_CNN 339 model on the task of classifying encrypted communication protocol, this article uses the preprocessed dataset to conduct experiments, and Table 3 displays the accuracy, retrieval, and F1-scores for the eight

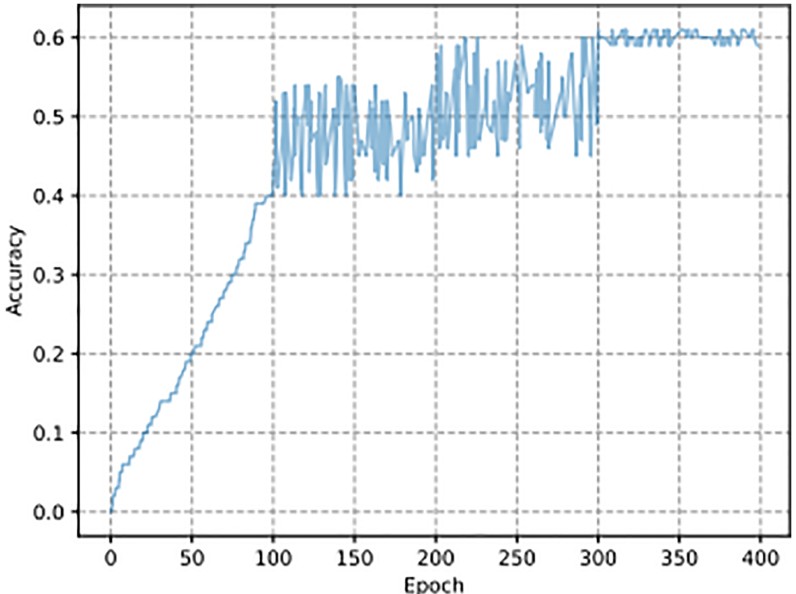

**Figure 5 SCGM plaintext guessing experiment results.**

**Table 3 Transformer-Attention CNN classification results.**

| Type of protocols | Accuracy | Retrieval | F1-score |
|---|---|---|---|
| International SSL | 97.32% | 97.95% | 97.67% |
| Secure socket layer | 98.16% | 97.51% | 97.83% |
| International IPSec | 97.72% | 97.36% | 97.54% |
| Secure socket layer | 96.58% | 97.51% | 97.58% |
| SSH | 98.93% | 98.81% | 98.87% |
| HTTP | 98.87% | 98.55% | 98.71% |
| DNS | 99.12% | 98.82% | 98.96% |
| UDP | 98.47% | 98.94% | 98.72% |

categories of traffic agreement. The classification results demonstrate that all protocols within each category achieve over 97% accuracy. Notably, unencrypted communication protocols such as HTTP, DNS, and UDP broadly exhibit greater 344 accuracy indices than encrypted protocols 345.

The promising cryptographic performance of the SCGM model requires evaluating how easily it may be vulnerable to adversarial attacks. Attacks from malicious entities become possible if they detect the model's weaknesses or structure because they can generate adversarial examples to trick the system. The ongoing research will concentrate on advancing model resilience because adversarial robustness functions are absent from their current state. We will add both adversarial training with tailored adversarial inputs and model regularization techniques during the training process to make the model resistant to potential attacks. The model needs robust optimization methods to build defensive

**Table 4 The experimental results of each model.**

| Model | Accuracy |
| --- | --- |
| 1D-CNN | 91.61% |
| Prt-CNN | 94.67% |
| Our model | 97.23% |

architectures because this enhances its ability to resist adversarial attacks. To verify the reliability and validity of the proposed model scheme, 346 subsequent comparative experiments will be conducted using the same dataset. The data in dataset 347 consists of raw traffic that has not been preprocessed, allowing for preprocessing based on the requirements of each model. The proposed model will be compared in the experiments with the one-dimensional convolutional neural network model proposed in the reference *Li et al. (2020)* and *Feng et al.*'s *(2020)* PrtCNN model. These two comparative models are representative in the field of protocol identification. The comparison results of each model are shown in 352 (Table 4). These findings did not separate the attention mechanism's impact on accuracy outcomes from the rest of the model since the mechanism was integrated to enhance feature concentration. An ablation study will assess the model's performance without the attention mechanism to measure its impact relative to the complete model. The ablation study will help calculate the attention mechanism's dedicated effect and provide critical knowledge about its role in accuracy enhancement.

Table 4 shows that the proposed model outperforms the other two models regarding various metrics, including recognition accuracy, with the 1D-CNN scheme exhibiting poorer performance. The proposed protocol identification model achieves an average recognition accuracy of 97.23%, which is 5.62% higher than the 1D-CNN scheme and 2.56% higher than the PrtCNN scheme. The proposed model utilizes the self-attention mechanism of the Transformer and the characteristics of Attention_CNN to filter out irrelevant features further, highlight important ones, and explore potential correlations from the perspective of spatiotemporal feature fusion. The SCGM model was tested against two established models: the 1D-CNN and the PrtCNN models. The 1D-CNN is a basic convolutional model designed for feature extraction from one-dimensional data, while the PrtCNN model has been widely used for encrypted traffic classification. Our results show that the SCGM model outperforms the 1D-CNN and PrtCNN models, with a recognition accuracy of 97.23%.

In comparison, the 1D-CNN model achieved an accuracy of 91.61%, and the PrtCNN model reached 94.67%. The SCGM model also demonstrated higher precision, recall, and F1 scores, further underscoring its ability to identify cryptographic algorithms and classify encrypted traffic protocols. The advantage of the SCGM model lies in its combination of convolutional neural networks with a self-attention mechanism, which allows it to focus on the most relevant features of encrypted data and adapt more effectively to varying encryption schemes.

Research on broadening generalization capacity remains essential to extending the proposed models across various cryptographic systems, including hybrid cryptosystems

and encryption features that may change. The deep learning models achieve data pattern recognition through convolutional networks and attention mechanisms, which indicates they can properly adapt to encryption systems that combine symmetric and asymmetric techniques. To establish total robustness and scalability, additional testing must be done on hybrid cryptosystems and evolving traffic data, including encryption protocol changes. To address the scalability issues, future work will concentrate on implementing dynamic batch size, which will enable the system to automatically adjust the batch size according to available computational resources and dataset features. The model's efficiency in handling larger datasets remains ensured when there is no memory or computational power overload. The model will be modified to process messages with adjustable lengths in addition to fixed key inputs. Because of the model's adaptable structure, different encryption schemes and real-world traffic conditions can be adequately handled.

Hybrid cryptographic frameworks, such as those used in modern secure communication protocols (*e.g.*, SSL/TLS), combine the efficiency of symmetric encryption for data encryption with the security of asymmetric encryption for key exchange. The SCGM model can be adapted to classify encrypted data from such hybrid systems, provided appropriate datasets are available for training. Future research will extend the model to handle these more complex encryption schemes, incorporating symmetric and asymmetric elements in the analysis. This will require updates to the model's architecture to accommodate the unique features of hybrid systems, such as key exchange processes and different encryption layer behaviors.

## CONCLUSIONS

This article proposes a multi-module combined CNN model and applies it to four symmetric encryption algorithms. Through experimental analysis, we demonstrate that the accuracy of predicting plaintexts improves as the number of training iterations increases, eventually stabilizing at a specific convergence point.

Additionally, we focus on the protocol identification of encrypted traffic data, incorporating appropriate preprocessing techniques. By analyzing the variability of captured network traffic, we employ session-level processing to enhance the accuracy of communication data identification. Furthermore, we examine the impact of individual temporal and spatial features on recognition performance and propose an algorithmic scheme that integrates both feature types to improve the identification of encrypted traffic protocols.

Passwords serve as a fundamental authentication mechanism, yet biometrics may eventually replace them. Our study of password security aims to prevent unauthorized access for personal gain and enhance data protection through precise analysis and evaluation. The key contributions of this research include improving the tracking of cryptographic algorithms and encrypted traffic protocols, thereby strengthening security across communication networks. This advancement is particularly relevant for Internet of Things (IoT) devices, financial systems, and cloud services applications, where accurate

encrypted traffic analysis is crucial. The model's adaptability to different encryption algorithms makes it a valuable tool for future cryptographic research and implementation.

Neural networks play a significant role in mapping relationships within data. Under optimal conditions, they can recognize and generalize underlying patterns, potentially aiding in cryptographic analysis. However, several aspects require further investigation, including selecting activation functions, configuring convolutional and pooling layers, and the overall architecture of fully connected layers. Future research should focus on refining these parameters to enhance model compatibility with different encryption schemes. Expanding data collection and analysis will also help build a more comprehensive and robust model.

While these models can enhance cryptographic research and security, they also pose risks, such as unauthorized decryption, privacy violations, and potential cyberattack misuse. Ethical concerns regarding surveillance and unauthorized data analysis must be addressed. To mitigate these risks, we emphasize the importance of responsible use in regulated environments, ensuring transparency, accountability, and adherence to ethical guidelines.

The current model employs a single 1D convolutional layer for feature extraction. Future research should explore deeper architectures by stacking multiple convolutional layers with residual blocks to enhance pattern recognition in cryptographic data. Additionally, integrating novel techniques with the model could further improve its ability to identify cryptographic patterns. Our research will continue to explore advanced network architectures to enhance accuracy and stability. Future developments will also focus on creating adaptive methods that dynamically adjust to evolving traffic patterns and emerging cryptographic techniques.

### Funding
This work is supported by the State Grid Beijing Electric Power Company Science and Technology Project "Research on Security Evaluation Techniques of Cryptographic Applications for Data Circulation" (Grant No. 52023023000A). The funders had no role in study design, data collection and analysis, decision to publish, or preparation of the manuscript.

### Grant Disclosures
The following grant information was disclosed by the authors:
State Grid Beijing Electric Power Company Science and Technology Project "Research on Security Evaluation Techniques of Cryptographic Applications for Data Circulation": 52023023000A.

### Competing Interests
Jiakai Hao, Ming Jin, and Yuting Li are employed with State Grid Beijing Electric Power Company, Beijing, China. The authors declare that they have no competing interests.

## Author Contributions

- Jiakai Hao conceived and designed the experiments, performed the experiments, analyzed the data, performed the computation work, prepared figures and/or tables, authored or reviewed drafts of the article, and approved the final draft.
- Ming Jin conceived and designed the experiments, analyzed the data, performed the computation work, authored or reviewed drafts of the article, and approved the final draft.
- Yuting Li performed the experiments, analyzed the data, performed the computation work, authored or reviewed drafts of the article, and approved the final draft.
- Yuxin Yang conceived and designed the experiments, performed the experiments, performed the computation work, prepared figures and/or tables, and approved the final draft.

## Data Availability

The code and raw data are available in the Supplemental Files and at Zenodo: Yuxin, Y. (2025). Neural Network-based symmetric encryption algo-rithm with encrypted traffic protocol identification [Data set]. Zenodo. https://doi.org/10.5281/zenodo.14856544.

The original data is available at GitHub: https://github.com/WillKoehrsen/recurrent-neural-networks/blob/master/data/old/patent_search/gp-search-deep-neural-networks.csv.

## Supplemental Information

Supplemental information for this article can be found online at http://dx.doi.org/10.7717/peerj-cs.2750#supplemental-information.

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
