# Peer review of "Neural network-based symmetric encryption algorithm with encrypted traffic protocol identification"

_PeerJ Computer Science, doi:10.7717/peerj-cs.2750_

## Round 0.1 · original submission · Major Revisions

dear authors,
thank you for your submission to our esteemed journal. Your manuscript has been read with interest by me and the experts in the field. I would like to inform you that the experts has pointed out a number of improvement areas where your manuscript needs improvement before we consider it further. Therefore, we advise you to carefully revise the manuscript and update it in light of reviewer comments and resubmit. please also consider the following points along side with the reviewer comments.

1. improve the language of the manuscript.
2. figures quality improvement needed.
3. write the potential benefits of your research.

thank you

·

Basic reporting

- Figures lack detailed captions explaining the experimental setups or the interpretation of results (e.g., Figures 1–4).

- While experimental outcomes are presented, there are no theoretical proofs or formal justifications for the model's performance. Including derivations or discussions of algorithmic robustness would strengthen the claims.

Experimental design

- The conversion of one-dimensional traffic data into two-dimensional grayscale images is mentioned, but there is no explanation of how critical parameters such as image resolution, padding strategies, or scaling are selected. Additionally, the criteria for choosing 28×28 feature maps are not justified, leaving the preprocessing methodology unclear.

- The manuscript briefly mentions using PyTorch and PyCharm for implementation but omits critical details about the environment (e.g., GPU specifications, dataset size, and training duration). Without this, replicating the study’s experimental conditions becomes challenging.

- The manuscript mentions that the parallel structure of the recognition scheme improves scalability, but no quantitative results (e.g., training time reduction, computational cost analysis) are presented to support this claim.

Validity of the findings

- The models are tested on four symmetric cryptographic algorithms and eight traffic protocol categories. How well would the proposed methods generalize to a wider range of algorithms or scenarios, such as hybrid cryptosystems or encrypted traffic with evolving features?

Additional comments

This study introduces a multi-module neural network model to analyze cryptographic algorithms and encrypted traffic protocols. It proposes a convolutional model for symmetric encryption and a spatiotemporal feature fusion model for traffic protocol identification. Results demonstrate high accuracy in classifying encrypted communication, with practical implications for improving cryptographic security. However, the paper suffers from the limitations listed below, which must be “fully” addressed before its reconsideration:

1- The SCGM model stabilizes at a hit rate of 60% for plaintext guessing. How do you justify this level of accuracy as “sufficient” for practical applications?

2- How representative is the dataset of real-world encrypted traffic, given that experimental results heavily rely on synthetic datasets? For instance, does the dataset include encryption schemes beyond the four specified algorithms?

3- In conv1d(input_, filter_shape, stride), your code implements a single 1D convolution operation. Why was no attempt made to stack multiple layers or integrate advanced architectures (e.g., residual blocks) to improve feature extraction for cryptographic patterns?

4- How do the results address potential adversarial attacks on the SCGM model, especially when the attacker knows its structure or weaknesses?

5- The inclusion of an attention mechanism for CNN layers improves feature focus, but its contribution to accuracy is not isolated in the results. Did the authors conduct an ablation study to quantify this specific impact?

6- How does the inclusion of pseudo-explicit text impact the generalizability of results to scenarios where the pseudo-text is absent?

7- Figure 1: The diagram lacks a detailed description of the components

8- The fixed batch_size = 512 and static key/message lengths (crypto_msg_len = 16) may limit scalability. How do you address the scalability challenges?

9- Figure 5: The accuracy curve stabilizes at 60%, but the axis labels/ legend do not indicate the specific training parameters or iterations, reducing interpretability.

Reviewer 2 ·

Basic reporting

This research explores the critical role of cryptography in power grid security, focusing on identifying cryptographic algorithms to enhance resistance against attacks. It introduces the SCGN model, which leverages convolutional neural networks to evaluate plaintext guessing capabilities for four symmetric encryption algorithms, assessing the model's learning potential and practical application. Overall, the paper seems good, and the flow is understandable. However, I have several suggestions for the authors to incorporate.

Experimental design

Provide detailed information about the datasets used for training and testing the SCGN model and protocol identification scheme.

Elaborate on the experimental setup for symmetric and asymmetric algorithm evaluations, including the metrics used to judge effectiveness.

Compare the SCGN model's performance with other existing cryptographic algorithm identification methods.

Validity of the findings

Discuss the limitations of the SCGN model and protocol identification scheme, such as scalability, adaptability to evolving encryption etc.

Propose strategies for future enhancements

Discuss whether the methodology can generalize to other symmetric and asymmetric encryption methods or emerging cryptographic standards.

Additional comments

Include examples of potential real-world scenarios where the proposed models could enhance power grid communication security.

Provide more detailed metrics, such as accuracy, precision, recall, and F1-score, to evaluate the models’ effectiveness comprehensively.
Extend the discussion to include hybrid cryptographic frameworks that combine symmetric and asymmetric encryption
Address any ethical implications or risks of using such models

---

## Round 0.2 · accepted · Accept

Based on the input on revised version of the paper, I'm pleased to inform you that the reviewers are now satisfied with the quality of the revised updated article. Therefore we are recommending it for publication. Congratulations

·

Basic reporting

The authors have addressed my comments; therefore, the paper can be accepted for publication in the present format.

Experimental design

The authors have addressed my comments; therefore, the paper can be accepted for publication in the present format.

Validity of the findings

The authors have addressed my comments; therefore, the paper can be accepted for publication in the present format.

Additional comments

The authors have addressed my comments; therefore, the paper can be accepted for publication in the present format.

Reviewer 2 ·

Basic reporting

No comment

Experimental design

No comment

Validity of the findings

No comment

Additional comments

No comment